# Direct observation of photoinduced sequential spin transition in a halogen-bonded hybrid system by complementary ultrafast optical and electron probes

Yifeng Jiang [1] ✉, Stuart Hayes [2], Simon Bittmann[3], Antoine Sarracini[2,9], Lai Chung Liu [4], Henrike M. Müller-Werkmeister [5], Atsuhiro Miyawaki[6], Masaki Hada [7], Shinnosuke Nakano[8], Ryoya Takahashi[8], Samiran Banu [8], Shin-ya Koshihara [8], Kazuyuki Takahashi [6], Tadahiko Ishikawa [8] ✉ & R. J. Dwayne Miller [2] ✉

A detailed understanding of the ultrafast dynamics of halogen-bonded materials is desired for designing supramolecular materials and tuning various electronic properties by external stimuli. Here, a prototypical halogen-bonded multifunctional material containing spin crossover (SCO) cations and paramagnetic radical anions is studied as a model system of photo-switchable SCO hybrid systems using ultrafast electron diffraction and two complementary optical spectroscopic techniques. Our results reveal a sequential dynamics from SCO to radical dimer softening, uncovering a key transient intermediate state. In combination with quantum chemistry calculations, we demonstrate the presence of halogen bonds in the low- and high-temperature phases and propose their role during the photoinduced sequential dynamics, underscoring the significance of exploring ultrafast dynamics. Our research highlights the promising utility of halogen bonds in finely tuning functional properties across diverse photoactive multifunctional materials.

Intermolecular interactions are essential for molecular recognition and functional materials with broad applications in crystal engineering, supramolecular chemistry, and drug design[1–3]. Halogen bonds are noncovalent directional intermolecular interactions like hydrogen bonds, but have been discovered relatively recently and studied less extensively until now. They arise from the attraction between halogen atoms with an electron-depleted σ-hole and nucleophilic sites on adjacent molecules. The distinctive nature of directional interactions is crucial for both crystal design and the photoinduced structural deformation, serving as a key element in the development of innovative photo-functionality. However, little is known about the role of the halogen bonds in photoinduced dynamics of supramolecular systems on the ultrafast timescale. A better understanding will facilitate the design of multifunctional materials that exploit synergistic

[1]European XFEL, Holzkoppel 4, 22869 Schenefeld, Germany. [2]Departments of Chemistry and Physics, University of Toronto, 80 St. George St., Toronto M5S 3H6 ON, Canada. [3]Max Planck Institute for the Structure and Dynamics of Matter, Luruper Chaussee 149, 22761 Hamburg, Germany. [4]Uncharted Software, 600-2 Berkeley St., Toronto M5A 4J5 ON, Canada. [5]Institute of Chemistry, University of Potsdam, Karl-Liebknecht-Str. 24-25, 14476 Potsdam-Golm, Germany. [6]Department of Chemistry, Graduate School of Science, Kobe University, 1-1, Rokkodai-cho, Nada-ku, Kobe, Hyogo 657-8501, Japan. [7]Tsukuba Research Center for Energy Materials Science, University of Tsukuba, 1-1-1, Tennodai, Tsukuba, Ibaraki 305-8573, Japan. [8]Department of Chemistry, School of Science, Tokyo Institute of Technology, 2-12-1 Ookayama, Meguro-ku, Tokyo 152-8551, Japan. [9]Present address: Paul Scherrer Institut, Forschungsstrasse 111, 5232 Villigen PSI, Switzerland. ✉e-mail: yifeng.jiang@xfel.eu; ktaka@crystal.kobe-u.ac.jp; tishi@chem.titech.ac.jp; dmiller@lphys2.chem.utoronto.ca

interactions with diverse properties, thereby expanding the range of materials for various applications.

Spin crossover (SCO) refers to a phenomenon observed in some transition-metal coordination complexes whereby a spin transition between the low-spin (LS) and high-spin (HS) states is triggered through changes in pressure, temperature, magnetic fields, or light irradiation[4–6]. Photoinduced SCO dynamics have been intensively studied using multiple methods[7–17] and some consensus about the general pathway of the photoinduced dynamics has been achieved in Fe(III) complexes (See Supplementary Discussion 1). Furthermore, the development of multifunctional SCO complexes has been actively pursued to control other functionalities with the SCO phenomenon by the interaction between the ions. Designing such systems can be a new strategy to realize synthetic photo-functional materials. Some of the studies were known for the coupling of conductivity[18,19], ferroelectric property[20], magnetism[21–23], and optical properties[24,25] in such SCO hybrid crystals.

Among such Fe(III) SCO hybrid compounds, a prototypical halogen-bonded multifunctional system, $[Fe(Iqsal)_2][Ni(dmit)_2]\cdot CH_3CN\cdot H_2O$ (**1**) [Iqsal = 5-iodo-$N$-(8'-quinolyl)-salicylaldiminate, dmit = 1,3-dithiole-2-thione-4,5-dithiolate] exhibits a thermally induced SCO transition of the $[Fe(Iqsal)_2]^+$ cations [transition temperature ($T_c$) = 150 K], with a strong correlation of a spin-singlet formation due to the dimerization of the $[Ni(dmit)_2]^-$ anions[22]. Direct current magnetic susceptibility measurements of the photo-response of **1** indicated that a spin transition can be reached by the light-induced excited spin state trapping effect[26]. In this system, we expect to explore photoinduced dynamics associated with halogen bonds in the ultrafast timescale triggered by the well-known SCO dynamics as discussed later. A direct measurement of ultrafast structural dynamics in combination with ultrafast optical spectroscopies is required to reveal possible intermediates and the reaction pathway. Understanding these ultrafast dynamics helps to unravel the fundamental mechanisms that drive the synergistic spin transition in the halogen-bonded multifunctional hybrid crystal **1**.

The structure of **1** in the high-temperature (HT, $T > T_c$) phase is shown in Fig. 1a. It comprises $[Fe(Iqsal)_2]^+$ cations, which are Fe(III)-centered SCO molecules with two iodine-substituted ligands[27] and the $[Ni(dmit)_2]^-$ anions which are arranged in a one dimensional zigzag molecular array with alternating face-to-face and side-by-side manner[22]. The formation of the intermolecular iodine-sulfur (I···S) halogen bond interaction in the HT phase and its competition with the π-stacking interaction within $[Ni(dmit)_2]^-$ anions were reported to be crucial in the thermally induced synergistic spin transition and controlling the arrangement and spin state of the $[Ni(dmit)_2]^-$ anions. Figure 1b shows the spin state of the $[Fe(Iqsal)_2]^+$ cation and the dimerization of $[Ni(dmit)_2]^-$ anions in HT and low-temperature (LT, $T < T_c$) phases[22]. The LT phase of **1** is characterized by the low-spin (LS) state of the $[Fe(Iqsal)_2]^+$ cation with strong dimerization (SD) of the $[Ni(dmit)_2]^-$ anions. In the HT phase, it exhibits the high-spin (HS) state of the $[Fe(Iqsal)_2]^+$ cation with weak dimerization (WD) of the $[Ni(dmit)_2]^-$ anions. These changes in spin state are accompanied by changes in the intra- and inter-molecular geometries. For the Fe-centered SCO, Fe-ligand expansion (0.10 Å) is the key feature, while a symmetric slipping displacement along the long molecular axis (0.89 Å) is the most significant mode influencing the dimerization of $[Ni(dmit)_2]^-$ anions. With these changes from the LT to the HT phase, the I···S distance related to halogen bond is shortened by 0.15 Å[22].

Here, we exploited a combination of three complementary methods under similar excitation conditions (Methods), including time-resolved transient absorption spectroscopy (TA), time-resolved mid-infrared vibrational spectroscopy (MIR), and ultrafast electron diffraction (UED), as schematically illustrated in Fig. 1c, investigating ultrafast electronic dynamics as well as mapping the structural reorganization of **1**. The results show sequential dynamics starting with

photoinduced SCO in the $[Fe(Iqsal)_2]^+$ cation followed by radical dimer softening of the $[Ni(dmit)_2]^-$ anions, and unveil a key transient intermediate state (TIS). We discuss the role of halogen bonds during this photoinduced sequential dynamics, and suggest the importance of investigating ultrafast dynamics.

## Results

The samples were cooled and maintained in the LT phase (Supplementary Method 1) as the ground-state reference and excited using a 400 nm laser pulse with $E\|c$ polarization. This should selectively excite the $[Fe(Iqsal)_2]^+$ cation from the LS ground state to the ligand-to-metal-charge-transfer (LMCT) state (Supplementary Method 2). The optical absorbance of the $[Ni(dmit)_2]^-$ anions should be small with $E\|c$ at this wavelength so that the changes of the $[Ni(dmit)_2]^-$ anions will be indirect in response to changes in the $[Fe(Iqsal)_2]^+$ cation. The pump fluence in this study is below the sample damage threshold, and peak power is within the single-photon excitation regime (Methods and Supplementary Method 3).

### Time-resolved transient absorption spectroscopy

In the TA measurement, time-resolved optical density (OD) changes in visible spectra (VIS) (415–650 nm) were measured to investigate the excited electronic state dynamics (Fig. 2). To facilitate the assignment of the transient spectra, ground-state absorption spectra of the LT and HT phases were obtained at 77 and 290 K, respectively. The difference spectra between the LT and HT phases (Fig. 2a) show that the OD of the HT phase is lower than that of the LT phase at wavelengths shorter than $\lambda_0 = 560$ nm and higher at wavelengths longer than $\lambda_0$ in the visible region, with some bleaching in the near-infrared (NIR) end of the spectrum. These features are known spectroscopic fingerprints of SCO and have been observed in TA data of other Fe(III) SCO compounds[11,28].

Transient spectra at selected time delays are displayed in Fig. 2b, while Fig. 2c, d present the time-resolved changes in the OD at short-time (−2 – +6 ps) and long-time (−20 – +550 ps) temporal windows at selected wavelengths. Figure 2b shows an approximate match of all the transient spectra after +100 ps time delay with the characteristic of the HT phase from the temperature-dependent measurement. We employed a global fitting approach based on singular value decomposition (SVD)[12] and obtained distinct spectral components (Supplementary Method 4).

In Fig. 2c, the short-time kinetic traces show rapid changes within our time-resolution, ($0.08 \pm 0.01$ ps), due to the photoexcitation and a relaxation process ($0.16 \pm 0.05$ ps) immediately following the rapid changes. We attribute the rapid change to the formation of the LMCT state. The following relaxation can be attributed to an intersystem crossing (ISC) from the LMCT to the HS state[11,28]. The second sub-ps relaxation component ($0.81 \pm 0.09$ ps) can be assigned to an intramolecular vibrational energy redistribution (IVR) process[8,10–15]. On the long-time, two relaxation components are observed, with time constants of $13.4 \pm 0.4$ and $50.1 \pm 0.5$ ps. They correspond to two additional relaxation processes after the formation of the HS state and have not been previously observed in other SCO TA studies. Their decay-associated spectra (DAS) features (Supplementary Fig. 8) and time constants differ significantly from those of the HS→LS relaxation or thermally induced SCO caused by heat deposited from the pump pulse[12,28–30]. We postulate these two relaxation processes are electronic changes of the $[Fe(Iqsal)_2]^+$ cations that are correlated to the dimer softening of the $[Ni(dmit)_2]^-$ anions. However, the spectroscopic fingerprints of the dimer softening have not been identified in the VIS range.

### Time-resolved MIR

Time-resolved MIR spectra were measured in the energy range of the intramolecular vibration modes to investigate the electronic dynamics and local structural changes (Fig. 3). Two IR spectral features

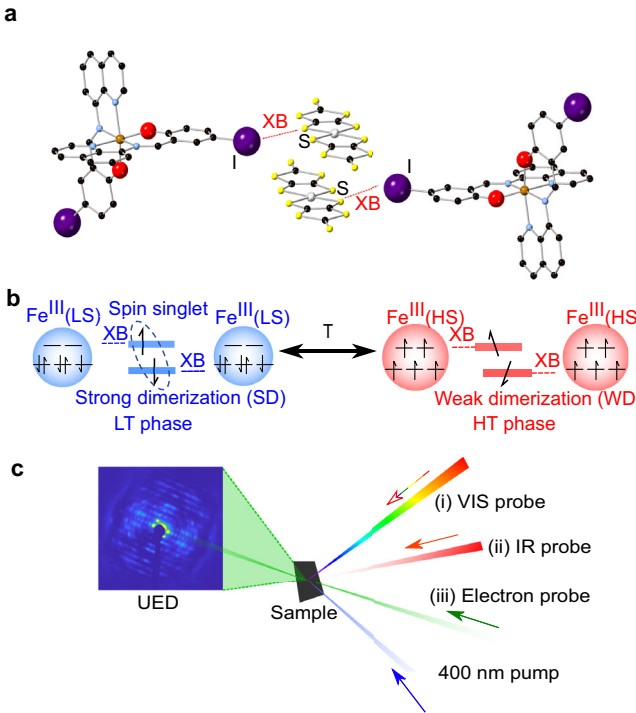

**Fig. 1 | Schematic structure, thermally induced phase transition, and experimental setup. a** Schematic of the structure of **1** in the high-temperature (HT) phase. Red dotted lines are the halogen bond (XB) between I and S atoms. Black balls are carbon atoms; blue balls are nitrogen atoms; red balls are oxygen atoms; brown balls are iron (III) atoms; purple balls are iodine atoms; yellow balls are sulfur atoms; white balls are nickel atoms. Hydrogen atoms are skipped to improve clarity. **b** Schematic of the local structure of **1** which shows the thermally induced phase transition. In the low-temperature (LT) phase (left panel), the system is in the low-spin (LS) state of $[Fe(Iqsal)_2]^+$ cations with singlet-spin state (strong dimerization (SD)) of $[Ni(dmit)_2]^-$ anions. It is in the high-spin (HS) state of $[Fe(Iqsal)_2]^+$ cations with paramagnetic state (weak dimerization (WD)) of $[Ni(dmit)_2]^-$ anions) in the HT phase (right panel). The blue and red circles represent $[Fe(Iqsal)_2]^+$ cations. Spin configuration on $t_{2g}$ and $e_g$ orbitals of the Fe(III) ion are shown inside the circles. The blue circles are smaller than the red circles to emphasize the Fe-ligand expansion during SCO. The horizontal thick lines between the $[Fe(Iqsal)_2]^+$ cations represent positions of the $[Ni(dmit)_2]^-$ anions. The arrows on the $[Ni(dmit)_2]^-$ anions represents the localized spins and become spin-singlet due to SD in the LT phase. **c** Schematic of the experimental setup and the diffraction images. After initial photoexcitation with 400-nm pump laser, ultrafast dynamics of **1** was measured individually by (i) transient absorption spectroscopy, (ii) IR reflectivity spectroscopy, and (iii) ultrafast electron diffraction.

at ~1450 and 1350 $cm^{-1}$ are assigned to a coupled stretching mode of C-C bonds in salicylaldimine rings of the $[Fe(Iqsal)_2]^+$ cation ligands (Supplementary Movies 1 and 2) and a C=C stretching mode of the $[Ni(dmit)_2]^-$ anions[31–34] (see inset of Fig. 3d) (see Supplementary Discussion 2 for assignments). These vibrational modes provide individual information on the dynamics of the $[Fe(Iqsal)_2]^+$ cations and $[Ni(dmit)_2]^-$ anions.

Figure 3a, b shows the relative transient changes of the IR reflectivity spectrum (ΔR/R) at selected time delays. In Fig. 3a, the transient spectrum at around 1450 $cm^{-1}$ ($[Fe(Iqsal)_2]^+$ mode) at +1 ps (blue curve) approximately matches the thermally induced one (black curve) (Fig. 3a), considering slight changes due to the non-equilibrium transient nature. Therefore, the fast changes of ΔR/R at +1 ps can be assigned to SCO in the $[Fe(Iqsal)_2]^+$ cation. The parameters of the C-C stretching mode of the Iqsal ligands (Supplementary Data Mov. 1 and 2) are sensitive to both the spin state of the Fe(III) center and the local structural change in the ligands. The fast change at +1 ps after photoexcitation is SCO in the $[Fe(Iqsal)_2]^+$ cation, and the

biexponential relaxation process after the fast change matches with slow dynamics observed in the above TA experiment.

In Fig. 3b, the ΔR/R spectrum at +1 ps displays a double-bottom feature, which is different from the thermally induced ΔR/R, which reflects the formation of the WD state of the $[Ni(dmit)_2]^-$ anions. A clear double-bottom shape in the reflectivity spectrum at around 1350 $cm^{-1}$ in Supplementary Fig. 12a is assigned to the C=C stretching mode of $[Ni(dmit)_2]^-$ anion. The peak position of this mode corresponds to the degree of dimerization of the paired $[Ni(dmit)_2]^-$ anions, and it shifts at $T_c$. The spectral change serves as a characteristic MIR fingerprint of the SD/WD state. The ΔR/R spectrum at +1 ps indicates a decrease of ε infinity value in the Lorentz model, which is attributed to the SCO related spectral change in the higher wavenumber range. Thus, the spectral change after initial photoexcitation is not related to the dimerization change, and $[Ni(dmit)_2]^-$ anions are still in the SD state at +1 ps. The ΔR/R spectrum at +10 and +100 ps show a gradual disappearance of the double-bottom feature. The spectrum at +100 ps is approaching the thermally induced phase transition ΔR/R spectrum, providing a spectral signature in agreement with formation of the WD state. In Fig. 3c, d, ΔR/R show rapid changes within 0.7 ± 1.2 ps (limited by our temporal resolution) after the photoexcitation and its relaxation with the biexponential character of 9.4 ± 1.1 and 45.8 ± 6.5 ps (Supplementary Method 4) matching with slow dynamics observed in the TA experiment. There is perturbed free induction decay in the negative time range, unrelated to the photoinduced dynamics[35]. The fitting results show the vibrational modes of the $[Fe(Iqsal)_2]^+$ cation and the $[Ni(dmit)_2]^-$ anion exhibit the same set of relaxation time constants, suggesting a close intermolecular coupling. Therefore, we observed that dimer softening from the SD to the WD states is a sequential slow process following the initial ultrafast SCO.

## Ultrafast electron diffraction

UED measurement can directly determine atomic positions in crystals with femtosecond temporal resolution and high structural sensitivity[36–41]. It is complementary to spectroscopic techniques, which measure transitions between energy levels, and offers unique insights into how atomic nuclei rearrange to stabilize spin-related electron distribution changes. A static electron diffraction pattern of **1** in the LT phase (125 K) is shown in Fig. 4a. Figure 4b shows the change in Bragg peak intensities due to thermally induced synergistic spin transition. This pattern is different to the photoinduced change at +5 ps (Fig. 4c). Still, it has an approximate match with transient change at +100 ps (Fig. 4d). Correction for the Debye-Waller (DW) effect was performed to eliminate its contribution (Supplementary Method 5). The time-dependent behaviors of some selected peaks in the short (−4 − +10 ps) and long (+10 − +100 ps) time delays are shown in Fig. 4e, with distinct structural changes in the fast and slow dynamics. For example, the time traces of intensity changes of Bragg peaks (0 4 0) and (0 3 1) display different behavior on different timescales: The intensity of peak (0 4 0) has a rapid decrease followed by small changes in the slow dynamics; the intensity of Bragg peak (0 3 1) consists of a rapid decrease followed by a gradual increase in intensity. In Fig. 4e, the plateau behavior between +2 and +10 ps implies a TIS that is different in structure from the ground state and the final HT-like excited state.

## Structural reconstruction

A reconstruction of the structural dynamics utilizing a parameterized molecular model[36] was conducted to determine the molecular motions. To reduce the number of degrees of freedom, key structural modes were selected to find the global best-fit structure based on the literature and our chemical intuition of structures and phase transitions[8,15,22,37]. These structural models allowed us to capture the most important features of the photoinduced dynamics while avoiding overfitting and chemically unreasonable structures. Following a careful preparation of the structural model (Supplementary Method 6),

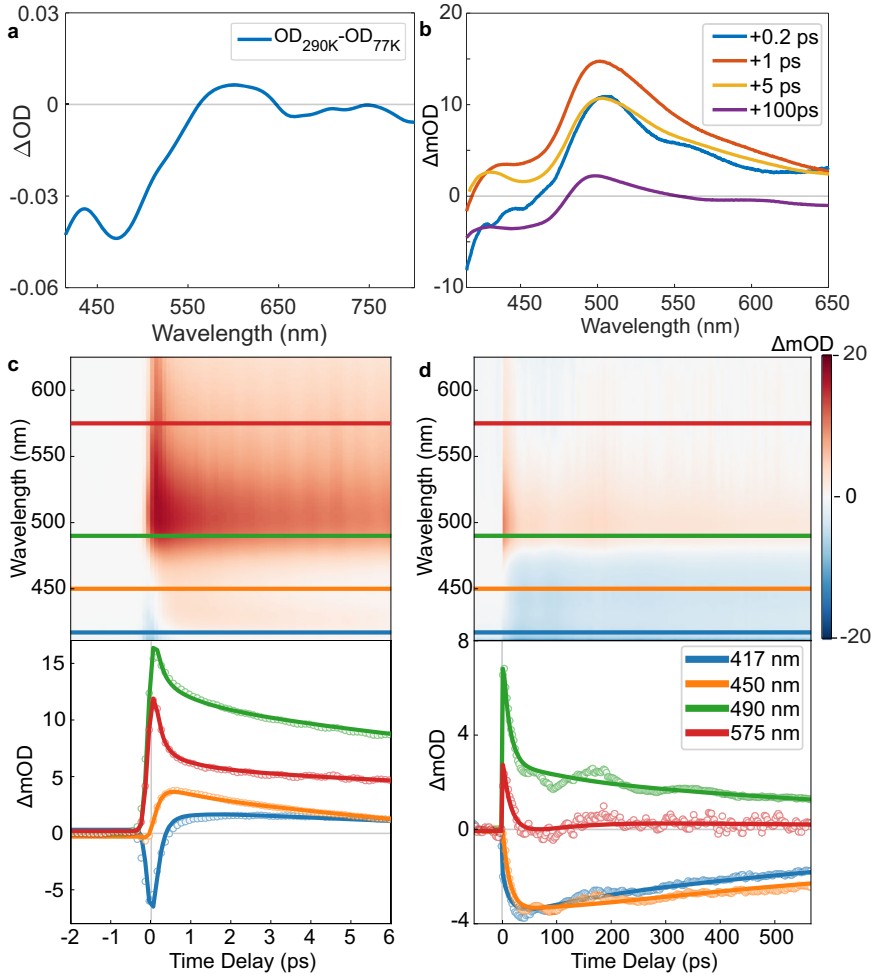

**Fig. 2 | Transmission VIS spectroscopy study. a** The differential optical density spectrum (ΔOD) between HT (290 K) and LT (77 K) phases. **b** Photoinduced ΔOD spectra at +0.2 (blue), +1 (red), +5 (yellow), and +100 ps (purple) measured at 77 K (LT phase). **c** Upper panel shows transient absorption scans in the visible region (contour plot), and lower panel shows temporal profiles at the specific wavelength between −2 and 6 ps (open circles). The simulated curves based on the exponential decay type functions (Supplementary Equation 1) are plotted as solid lines. **d** Upper panel shows transient absorption scans in the visible region, and lower panel shows temporal profiles at the specific wavelength between −20 and 550 ps (open circles). The simulated curves based on the exponential decay type functions are plotted as solid lines. In the upper panels of **c**, **d** the colored horizontal solid lines present the selected wavelengths in the lower panels of **c**, **d**. The specific wavelengths are designated by the same-colored symbols and lines in the lower panels **c**, **d**. Source data are provided as a Source Data file.

two key structural modes were set, namely the Fe-ligand expansion in the $[Fe(Iqsal)_2]^+$ cation ($\mathbf{p}_{SCO}$) and slipping displacement of the paired $[Ni(dmit)_2]^-$ anions in the direction of the molecular long axis ($\mathbf{p}_{dmit}$) to decompose atomic motions from the ground state (LS-SD) to the HT-like state (HS-WD) (Fig. 5a). The structural refinement algorithm determines structures of **1** at each time point by maximizing the correlation between the experimentally observed and simulated diffraction intensities (comprising 80 Bragg peaks) sampling the full parameter space between the ground and final states. The temporal evolutions of the Fe-ligand distance (average bond distance of Fe-N and Fe-O coordination bonds) and the displacement of the $[Ni(dmit)_2]^-$ anions (distance between two paired $[Ni(dmit)_2]^-$ anions projected along the molecular long axis) are displayed in Fig. 5b, c.

In addition to the structural analysis, the temporal response of the Bragg peaks was fitted in a similar way to the ultrafast spectroscopy, with a series of decaying exponentials using the global fitting based on SVD[12]. Four distinct components were found with time constants similar to those observed in the TA and MIR measurements, illustrating that both spectroscopic and UED measurements probe the same process. The time constant of the shortest relaxation process of the diffraction data ($0.5 \pm 0.2$ ps) matches the ISC processes observed in the TA data and other literature[11,13,15,28] (limited by the $(0.4 \pm 0.05)$-ps

instrument response time of our UED setup). This process can be attributed to an ultrafast Fe-ligand expansion associated with the formation of the HS state. The second $0.9 \pm 0.3$ ps component shows a further structural relaxation associated with the IVR process after the formation of the HS state[11,13,15,28]. The plateau behavior with the third $9.4 \pm 0.6$ ps component is corresponding to TIS. TIS is found to be comprised of the HS state of the $[Fe(Iqsal)_2]^+$ cations with SD of the $[Ni(dmit)_2]^-$ anions (HS-SD). The evaluated Fe-ligand expansion after the IVR process (0.06 Å) at +2 ps is still smaller than that found in the HT phase (0.09 Å) as shown in Fig. 5b. It suggests following the initial photoexcitation, the excited HS molecule undergoes structural rearrangement while constrained within a smaller LT unit cell and under the influence of chemical pressure[14,15,42,43]. The last slow ($54 \pm 6$ ps) component corresponds to an ensemble-averaged picture of the dimer softening mainly by a 0.90 Å slipping displacement of the $[Ni(dmit)_2]^-$ anions in the direction of the molecular long axis shown in Fig. 5c.

## Discussion

To clarify the role of halogen bonds in the dynamics of the photo-induced sequential spin transitions, we plotted the temporal profile of the I⋯S halogen bond distance in Fig. 5d after refining the structures of

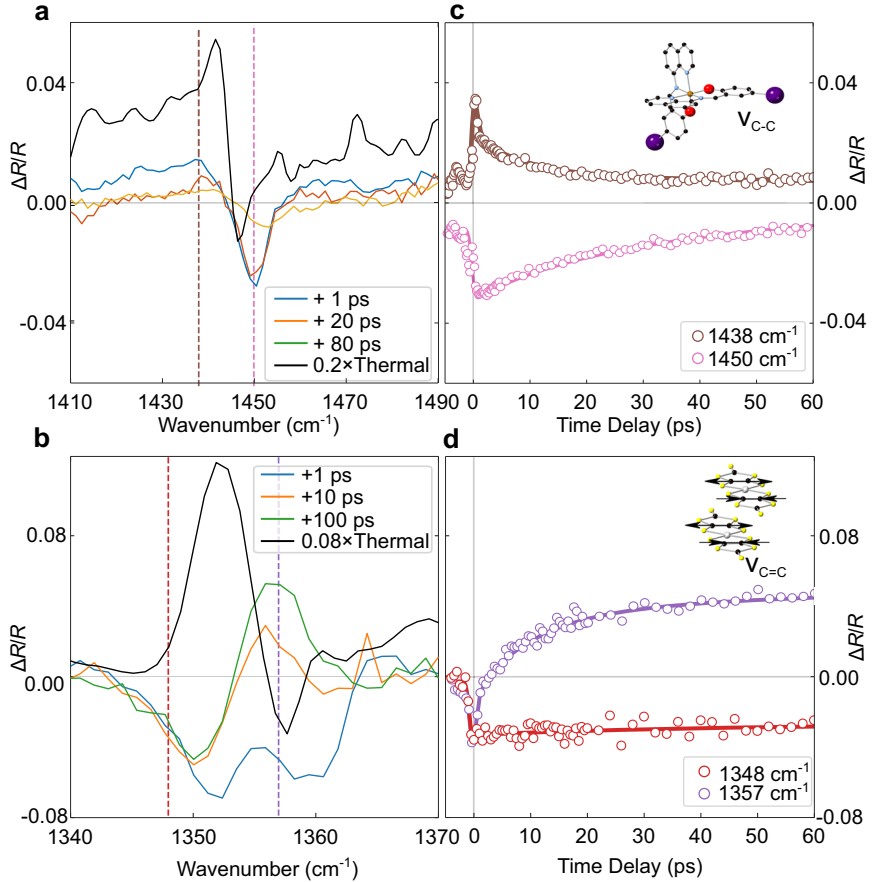

**Fig. 3 | MIR reflectivity study. a** Photoinduced relative change of the reflectivity spectra ($\Delta R/R$) from 1410 to 1490 cm$^{-1}$ (in which a coupled stretching mode of C-C bonds in salicylaldimine rings of the ligands of [Fe(Iqsal)$_2$]$^+$ cations are observed, see Supplementary Movies 1 and 2) at selected time delays measured at 100 K (LT phase). Scaled (0.2) thermally induced $\Delta R/R$ spectra between 290 and 90 K are also plotted. **b** Photoinduced $\Delta R/R$ spectra from 1340 to 1370 cm$^{-1}$ (in which a C=C stretching mode of the [Ni(dmit)$_2$]$^-$ anions are observed) at selected time delays. Scaled (0.08) thermally induced $\Delta R/R$ spectra between 290 and 90 K are also plotted. **a, b** the vertical dashed lines indicate the wavenumbers of the probe energies of the selected temporal profiles in **c, d**, respectively. **c** Temporal profiles of selected wavenumbers, 1438 and 1450 cm$^{-1}$. The inset shows the schematic molecular structure of [Fe(Iqsal)$_2$]$^+$ cations. The solid lines show the results of the global fitting (Supplementary Equation 1). **d** Temporal profiles of selected wavenumbers, 1348 and 1357 cm$^{-1}$ (open circles). The inset represents a C=C stretching mode of the [Ni(dmit)$_2$]$^-$ anions. The solid lines show the results of the global fitting. Source data are provided as a Source Data file.

**1** at each time delay with UED data. Additionally, the halogen bond interaction energies for the LT and HT phases as well as the refined time-resolved structures of **1** are quantified using the quantum theory of atoms in molecules (QTAIM)[44]. The analysis of the Laplacian of the electron density for the LT and HT phases in a thermal equilibrium reveals that a bond critical point (BCP) exists between the [Fe(Iqsal)$_2$]$^+$ cation and [Ni(dmit)$_2$]$^-$ anion (see Supplementary Fig. 13a, b). These findings indicate the mechanism of the synergistic spin transition is not a competition between the spin-singlet and halogen bond formations[22], but the synchronization between the two spin transitions through the halogen bond. Furthermore, when the QTAIM is adapted to the refined time-resolved structures of **1**, one BCP involved in the halogen bonding interaction is always present at each time delay (Supplementary Movie 3). These suggest that halogen bonding interactions play an important role in the photoinduced conformational changes, offering a better understanding of the dynamics underlying this synergistic spin transition.

With the present theoretical calculations based on the UED results, we propose a local vibrational energy transfer (VET) and strain process on the molecular scale, mediated mainly by the halogen bond, to drive the dimer softening of the [Ni(dmit)$_2$]$^-$ anions in 50 ps following SCO in the [Fe(Iqsal)$_2$]$^+$ cation. In Fig. 6, we conclude SCO and dimer softening processes occur along a curved trajectory on the potential energy surface. On the fast time delay, photoexcitation of the [Fe(Iqsal)$_2$]$^+$ cation drives the expansion of the SCO ligand shell and reaches TIS (HS-SD). The TIS is unstable and the photoexcited [Fe(Iqsal)$_2$]$^+$ cations must exchange their deposited excess energy with the surrounding environment via VET[45,46]. Among various intermolecular interactions, the halogen bond is one of the strongest interactions, with less than the sum of van der Waals (vdW) radii (3.78 Å). Because the VET process has been reported to be related to the strength of the intermolecular interactions[47,48], the halogen bond is likely to be one of the prime pathways for VET from the photoexcited [Fe(Iqsal)$_2$]$^+$ cations to their nearest-neighbor [Ni(dmit)$_2$]$^-$ anions. In Fig. 5c, the energy difference between the SD and WD states is -10 kJ/mol, which is 0.1 eV/molecule (Supplementary Method 7). The VET process from photoexcited SCO molecules (absorbed 3.09 eV) can provide enough energy to overcome the above energy difference to drive dimer softening of [Ni(dmit)$_2$]$^-$ anions. Another important fact is that the rapid expansion of the SCO ligand shell builds strain on the nearest-neighbor [Ni(dmit)$_2$]$^-$ anions in the direction of the I···S halogen bond. On the long-time, this additional strain proceeds with the re-establishment of the structural equilibrium along with the dimer softening of the neighboring [Ni(dmit)$_2$]$^-$ anions as a relaxation process associated with the shortening of the halogen bond (Fig. 5d). Recent works show similar timescale dynamics related to the photoinduced phase transition due to the elastic amplification induced by the volume expansion of the crystal[28,49,50]. However, it is worth

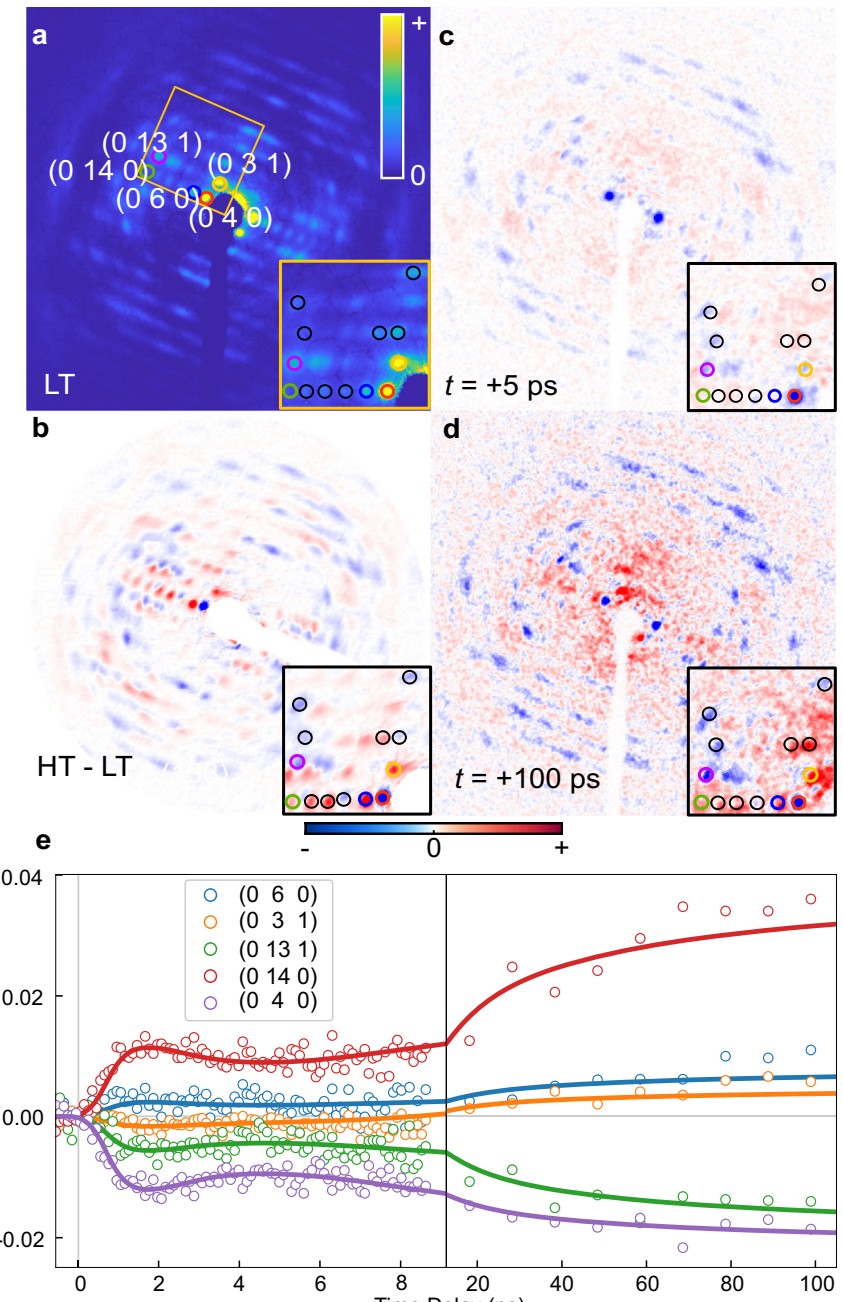

**Fig. 4 | Ultrafast electron diffraction study. a** Diffraction pattern measured at 125 K (LT phase) without photoexcitation. Color circles show the positions of the selected Bragg peaks. **b** Difference between the diffraction patterns of the 125 K (LT) and 300 K (HT). **c** Photoinduced signal measured after 5-ps time-delay at 125 K. **d** Photoinduced signal measured after 100-ps time-delay at 125 K. Insets of **a**–**d** show the selected area of the diffraction pattern indicated by the yellow rectangle in **a**. The black circles in the insets are the positions of the several selected Bragg peaks. **e** Kinetic traces of selected Bragg peaks' intensity. Short-time relative change in the intensity of selected Bragg peaks from −0.5 to 9 ps, and long-time relative changes in the intensity from +10 to 150 ps. The solid lines show the results of the global fitting (Supplementary Equation 1). Source data are provided as a Source Data file.

emphasizing that global volume expansion may be related to the dimer softening process, but it is unlikely to drive the phase transition on the timescale we have studied[28,49]. (Supplementary Discussion 3). Furthermore, the global volume expansion is not observed in the UED data (Supplementary Discussion 4).

In this study, we have extended the application of UED to a much larger multifunctional complex that revealed a sequential spin transition by direct observation of the nuclear reorganization (Supplementary Movie 4). This shows excellent agreement with the electronic dynamics from TA and the vibrational dynamics from MIR measurements, allowing us to assign the excited states involved, as shown in

Fig. 6 with great confidence. This study offers an ultrafast SCO on the Fe center on a sub-picosecond timescale producing a photoinduced TIS distinct from the ground and final excited states and sequential slow dynamics related to the dimer softening of [Ni(dmit)$_2$]$^-$ anions with a timescale of about 50 ps. We propose the initial local change in the Fe state initiates energy transfer and builds strain to the neighboring dimers, inferring the halogen bond is important in the structural relaxation process. Our study underscores the importance of ultrafast investigations that monitor electronic and structural dynamics following photoexcitation. It unveils the nature of ultrafast processes, providing insights with profound implications on a broader

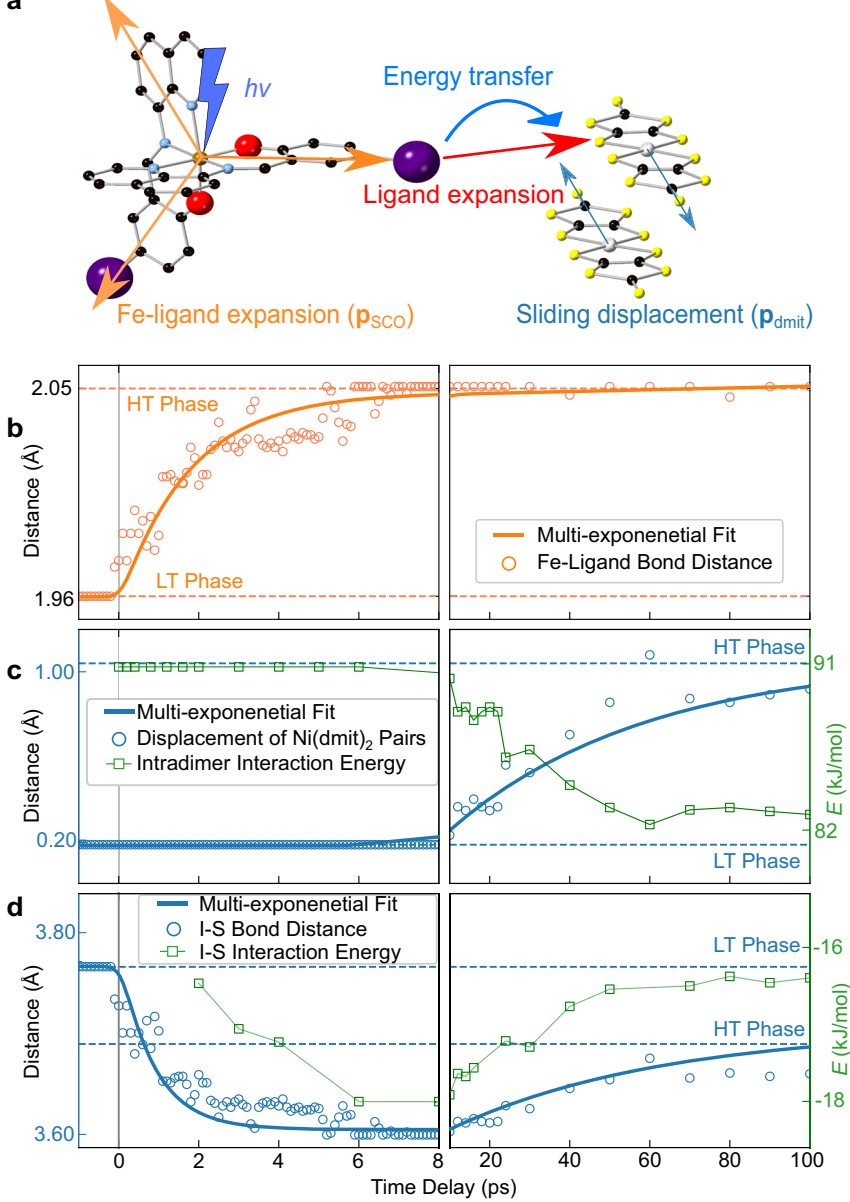

**Fig. 5 | Atomic motions and intermolecular interaction energy. a** Schematic ultrafast atomic motions of **1**, presented by two structural dynamics groups, Fe-ligand expansion in the molecules ($\mathbf{p}_{SCO}$) and slipping displacement of the [Ni(dmit)$_2$]$^-$ anions in the direction of the molecular long axis ($\mathbf{p}_{dmit}$). **b** Temporal evolution of the Fe-ligand distance (average bond distance of Fe-N and Fe-O coordination bonds) in short- and long-time-delay (open circles). **c** Temporal evolution of the displacement of the [Ni(dmit)$_2$]$^-$ anions (distance between two paired [Ni(dmit)$_2$]$^-$ anions along the molecular long axis, open circles) and

intradimer (between paired [Ni(dmit)$_2$]$^-$ anions) interaction energy (open rectangles) in short- and long-time-delay. **d** Temporal evolution of I⋯S Halogen bond distance (open circles) and halogen bond interaction energy (open rectangles) in short- and long-time-delay. **b**–**d** solid lines and horizontal dashed lines show the fitting curves using multi-exponential decay functions (Supplementary Equation 1) and values at HT (175 K) and LT (105 K) phases, respectively. Source data are provided as a Source Data file.

scale. Finally, we expect that halogen bonds can be utilized to achieve fine-tuned control over functional entities in other photo-active supramolecular systems with potential implications for developing fast-response optical data storage with multi-level cell capability.

## Methods
### Sample preparation
Single crystals of [Fe(Iqsal)$_2$][Ni(dmit)$_2$]·CH$_3$CN·H$_2$O (**1**) (Iqsal = 5-iodo-*N*-(8′-quinolyl)-salicylaldiminate, dmit = 1,3-dithiole-2-thione-4,5-dithiolate) were prepared using the same method as previously reported[22].

### Laser excitation conditions
In this study, we used similar laser excitation conditions for all three pump-probe experiments. At the sample position, the pulse duration of the pump laser was 60 fs for the TA and the UED and 90 fs for MIR, and the center wavelength was 400 nm. The incident excitation fluence was 0.55 mJ/cm$^2$ or 10.11 GW/cm$^2$ for the TA and the UED and 1.7 mJ/cm$^2$ for MIR. This excitation fluence enables hours of data collection without sample damage. (Supplementary Fig. 6). The polarization of the pump laser was set to *E*||*c* to only pump the iron (III) center in the [Fe(Iqsal)$_2$]$^+$ cation (Supplementary Method 2). From the crystal structure and absorption at 400 nm, the excitation fraction

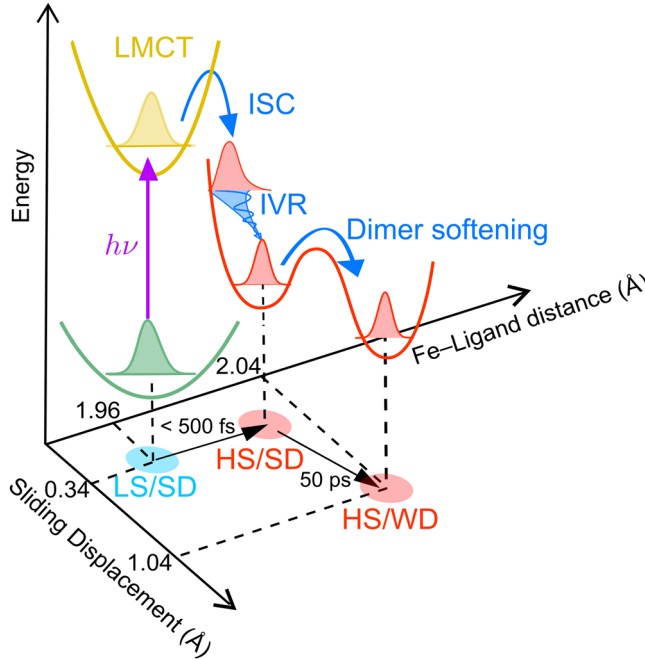

**Fig. 6 | Schematic of the proposed reaction pathway of the photoinduced spin transition of 1, indicated by the TA, MIR and UED experiments.** Initial photoexcitation (*hν*) of the ligand-to-metal-charge-transfer (LMCT) transition of the [Fe(Iqsal)$_2$]$^+$ cations in the LT phase (low-spin (LS) with strong dimerization (SD)) is followed by multiple intersystem crossing (ISC) processes and an intramolecular vibrational energy redistribution (IVR) process in the high-spin (HS) state within 0.8 ps. After 15 ps, the [Ni(dmit)$_2$]$^-$ anions start to rearrange from SD to weak dimerization (WD) states as a slow approximately 50 ps dimer softening dynamics.

(fractional population of the excited state) was calculated to be 7.1%. The excitation fraction was also calculated to be 5.4% from the change of the Bragg peak intensities at long time delay and the changes of the Bragg peak intensities during thermally induced phase transition[14,36], in good agreement with the previous calculation.

## Transient absorption experiment

The single crystal of the sample was ultramicrotomed to be 100 nm slices parallel to the *bc* plane, which is the most developed crystal face of the bulk single crystal and mounted on 1 mm thick UV-fused silica windows. The directions of the axes were judged from the sample shape and the anisotropy of the transmission spectra at room temperature. The visible probe light (380–700 nm) was produced by focusing circularly polarized 800 nm laser pulses in a 200 μm cuvette of water[51]. The repetition was set to be 125 Hz.

## MIR experiment

Both static and time-resolved optical measurements in the energy range of the intramolecular vibration mode, MIR range, were performed on a *bc* plane of a bulk single crystal using the reflection mode. The *b*-axis polarized static reflectivity spectra, and their temperature dependence in the MIR range were measured by a Fourier transform infrared (FT-IR) spectrometer equipped with a reflective objective. The *b*-axis polarized reflectivity change spectra *ΔR/R*(*t*) in the MIR range were measured by the pump-probe method using a 90 fs Ti:sapphire regenerative amplifier [fs-RGA, photon energy = 1.57 eV (792 nm), repetition rate = 1 kHz] as the light source. The MIR pulse that covers the wavenumber range = 1340–1500 cm$^{-1}$ for the probe pulse was generated by optical parametric amplification (OPA) and differential frequency generation (DFG) from the remaining output of the fs-RGA. The spectra of the reflected light from the sample were measured

using a grating-type monochromator. The synchronized optical chopper was inserted to reduce the repetition rate of the pump pulse to 500 Hz to measure the *ΔR/R*(*t*).

## UED experiment

The single crystal was ultramicrotomed to thin slices. The typical size of these slices was 300 μm × 300 μm × 100 nm. They were mounted on 400-mesh TEM grid with a layer of amorphous carbon. Experiments were performed at 125 K. The sample was oriented so that the electron beam was parallel to the crystal *a*-axis. A compact DC electron diffraction setup operating at 80 kV was used with minor modifications to the instrument[34]. The electron beam contained $1.6 \times 10^4$ electron per pulse with a spot size of (110 ± 10) μm FWHM at the sample position, and the electron pulse duration was about 300 fs FWHM. The repetition rate was 125 Hz, which allows the excited sample to relax back to the initial ground state before the arrival of next pump pulse as evidenced by the elimination of residual signal at negative time delays. For each time-delay, a total 25,000 pulses of electrons ($4 \times 10^8$ electrons) were used to accumulate sufficiently high signal-to-noise electron diffraction patterns.

## Quantum chemistry calculation

The geometry optimization and vibrational frequencies of the LS and HS states of the [Fe(Iqsal)$_2$]$^+$ cation were computed at the B3LYP functional[52,53] using the Wachters-Hay basis set for the Fe atom[54,55], the Stuttgart PLC ECP (SDD) basis set for the I atom[56], and the 6-31 G(d,p) basis set for the H, C, N, and O atoms[57] using the Gaussian 16 program package[58]. The atomic coordinates of the LS and HS [Fe(Iqsal)$_2$]$^+$ cations were taken from the literature[22]. The scaling factor for the frequencies of the vibrational mode is determined as 0.975 by comparing the computed spectrum with the experimentally observed one. Optimized atomic coordinates of LS state structure is provided as Supplementary Data 1. Optimized atomic coordinates of HS state structure is provided as Supplementary Data 2.

The electron density distributions involved in the halogen bond between the [Fe(Iqsal)$_2$]$^+$ cations and [Ni(dmit)$_2$]$^-$ anions were computed by single-point calculations using the M06 functional[59]. The Douglas-Kroll-Hess second-order scalar relativistic calculations requested relativistic core Hamiltonian were performed using the DZP-DKH basis set for all atoms[60,61]. The atomic coordinates of the [Fe(Iqsal)$_2$]$^+$ cation and [Ni(dmit)$_2$]$^-$ anion were taken from the UED structural data at each time delay. The topological analysis of the electron density distribution with the QTAIM[44] was carried out using the Multiwfn program[62]. The interaction energies of the halogen bond were estimated by the equation of binding energy for charged complexes in the literature[63].

## Structural parameters and Pearson correlation coefficient

Custom code has been uploaded (See Code availability), which was developed to build the parameter grid space to search the parameters and structures with the highest Pearson correlation coefficient, which is the best explanation for the observed data, at three key delay points: <0 ps (LS-SD) state, 6–8 ps (HS-SD) state, and 90–100 ps (HS-WD) state. Reduced UED data at these three key delay points have been uploaded in the capsule.

From the Pearson correlation coefficient calculation, we refined the molecular structures at different time delays and then obtained bond length information, which is shown in Fig. 4 of the paper. More details are explained in Supplementary Method 6 and Supplementary Fig. 10.

## Data availability

The processed data of transient absorption and MIR are available at the Source Data file. The processed UED data and crystal information data used for the structural refinement are available at Code Ocean

[https://codeocean.com/capsule/2959446/tree]. Source data are provided with this paper.

## Code availability

Custom code has been uploaded to Code Ocean [https://codeocean.com/capsule/2959446/tree], containing code used for the time-dependent model of atomic motions.

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

## Acknowledgements

This research was supported by the Natural Sciences and Engineering Research Council of Canada, Max Planck Society and the excellence cluster "The Hamburg Center for Ultrafast Imaging–Structure, Dynamics and Control of Matter at the Atomic Scale" of the Deutsche Forschungsgemeinschaft (R.J.D.M.), JSPS KAKENHI Grant Numbers JP18H05208 (M.H., S.K., and T.I.), JP19K05402 (K.T.), and JP21K03427 (T.I.). Y.J. acknowledges the financial support from European XFEL GmbH. We would like to thank Jordan Wenzell (Department of Chemistry, University of Toronto) for help in sample preparation.

## Author contributions

Y.J., L.L., M.H., K.T., T.I., and R.J.D.M. conceived the experiments. Si.B. and A.S. performed the transition absorption experiment, analyzed and interpreted the data with Y.J., L.L., K.T., T.I., and R.J.D.M. T.I., S.N., R.T., Sa.B., and S.K. performed the MIR vibrational spectroscopy experiment, analyzed, and interpreted the data with Y.J., S.H., L.L., H.M., K.T., and R.J.D.M. Y.J. and S.H. performed the UED experiment, analyzed and interpreted the data with L.L., Si.B., A.S., H.M., K.T., T.I., and R.J.D.M. A.M. and K.T. synthesized the single crystals. K.T. and T.I. performed the quantum chemistry calculation. S.H. prepared the samples for UED. All authors discussed the results and contributed to the writing of the manuscript.

## Funding

## Competing interests

The authors declare no competing interests.
