## [Peer Review File · Nature Communications]

Direct observation of photoinduced sequential spin transition in a halogen-bonded hybrid system by complementary ultrafast optical and electron probesEditorial Note: This manuscript has been previously reviewed at another journal. This document only contains reviewer comments and rebuttal letters for versions considered at *Nature Communications*. Mentions of prior referee reports have been redacted.

REVIEWER COMMENTS

Reviewer #1 (Remarks to the Author):

In this revision Jiang, Takahashi, Ishikawa, Miller and co-workers explore an iron(III) SCO material where dimerization of the $[\text{Ni}(\text{dmit})_2]$ - unit is coupled to spin crossover and mediated by halogen bonding. The authors explore this using an ultra-fast pump-probe system to explore the changes that occur in this multifunctional iron(III) SCO material.

Since the initial submission the authors have substantially revised the paper and have clearly tried to take on board the queries of myself and the other reviewers. I am particularly pleased to see that they have undertaken an additional spectroscopic study on an ultrathin sample that strongly suggests that the sample retains its composition despite being so thinly sliced.

However, there are still a few points to be addressed. In their response the authors state that they have calculated the %HS to be 7.1%, how was this number calculated?

The authors also indicate that they've created a new movie called extended data movie 3, but I can only see two movies. Have the authors forgotten to upload it or is this the new black and white Laplacian movie?

Minor points

1. Lines 59-60, the ligand is negatively charged and as per IUPAC rules it should be 5-iodo-N-(8'-quinolyl)-salicylaldiminate. This error is present throughout the manuscript.
2. Line 124, DAS is used but not explained.
3. Figure 1 remains more or less the same as before. Both myself and the other referee suggested that arrows or similar indicators be used to show the changes that occur upon photoexcitation.
4. A constant frustration for me in the ESI is having to move backwards and forwards to find the figures referred to in the supplementary text. As this is the ESI, it would be far better to place the figures near the text where they are referred to.
5. Extended figure 3, the caption use the word 'blu', when it should be blue.
6. In extended figure 12 it's very hard to read the labels for the atoms. I'd suggest that they are made larger or a different colour is used to make them more visible. This in fact applies to the new black and white movie they've created too.

Overall, I am happy with the changes made by the authors and I think the manuscript now more accurately reflects the results and clarifies the areas of confusion. Provided that they undertake these revisions I would be happy to recommend the paper for publication in *Nature Communications*.

Reviewer #2 (Remarks to the Author):

The paper is a new version of the paper by Yifeng Jiang et al, previously submitted to [REDACTED], is about ultra-fast pump-probe study of photoinduced phenomena in a material made of a photoactive spin crossover molecular unit, connected through halogen-bonds to $(\text{dmit})_2$ units exhibiting strong or weak dimerization. The authors used ultrafast optical and infra-red spectroscopies and complementary ultrafast electron diffraction, to gain knowledge on ultrafast and out-of-equilibrium photoinduced dynamics. The experimental data reveal ultrafast changes within 500 fs and slower dynamics within 50 ps. In this new version of the paper, as well as Supp.

Info, the authors have considered the most important remarks and have improved the presentation of the results. The analysis of the results and the discussion are more relevant and rigorous in the present version, with several additional discussions about the method or the data analysis, and the associated limitations, such as about the role of the volume change and the halogen bonds. The extended data, especially the one about Halogen bond, make the discussion stronger. Now it is clear why the two different timescales can be associated with two different dynamics and how they couple. This work is very promising for exploring enhanced control of multifunctional through halogen bond and it can attract a broad community of researchers.

The reviewer recommends publication of the paper in its present form, considering the minor points below:

1- It is indicated line 111 "Fig. 2b shows an approximate match of all the transient spectra of after +1 ps time delay with the characteristic of the formation of the HS state from the temperature-dependent measurement."

However, few lines above it is indicated "OD of the HT phase is lower than that of the LT phase at wavelengths shorter than $\lambda_0 = 530$ nm and higher at wavelengths longer than λ_0 in the visible region" Therefore, it seems from Fig2b that it is only after 100 ps that OD transient spectra represent an approximate match characteristic of the formation of the HS state."

I also found a typo: "transient spectra of after +1 ps time delay"

◇ "transient spectra after +1 ps time delay"

Reviewer #2 (Remarks on code availability):

Dear editor,

I'm sorry, I am not able to review the code. I have no electron data to run or test.

1 Authors thank the editor and referees for careful reading and various advice. Our great honor is
2 that the referee highly evaluated the revised manuscript and were satisfied with our point-to-
3 point response to their previous comments. Regarding the additional comments and minor
4 points raised by the referees, we, coauthors, believe all problems pointed out have been
5 addressed, and we apologize for the typo and inconsistency in our manuscript.

6 During considering the response to referees' comments, we noticed there was typos in Fig. 4f
7 and the labels are inconsistent with the main text. We corrected the typos, from ξ to p. Due to
8 this correction, inconsistency between main text and Fig. 4f disappeared.

9

10 Reviewer #1 (Remarks to the Author):

11 In this revision Jiang, Takahashi, Ishikawa, Miller and co-workers explore an iron(III) SCO
12 material where dimerization of the $[\text{Ni}(\text{dmit})_2]$ - unit is coupled to spin crossover and mediated
13 by halogen bonding. The authors explores this using an ultra-fast pump-probe system to explore
14 the changes that occur in this multifunctional iron(III) SCO material.

15 Since the initial submission the authors have substantially revised the paper and have clearly
16 tried to take on board the queries of myself and the other reviewers. I am particularly pleased
17 to see that they have undertaken an additional spectroscopic study on an ultrathin sample that
18 strongly suggests that the sample retains its composition despite being so thinly sliced.

19 However, there are still a few points to be addressed. In their response the authors state that
20 they have calculated the %HS to be 7.1%, how was this number calculated?

21 The %HS mentioned here is the excitation fraction (fractional population of the excited HS
22 state). As described in **Lines 288-292** of Method in the main manuscript, the excitation fraction
23 in this work was calculated by two complementary methods. First, the excitation fraction was
24 calculated from crystal structure and optical absorption at 400 nm. Then, the excitation fraction
25 was also calculated from the change of the Bragg peak intensities at a long-time delay and the
26 changes of the Bragg peak intensities during thermally induced phase transition.

27 The authors also indicate that they've created a new movie called extended data movie 3, but I
28 can only see two movies. Have the authors forgotten to upload it or is this the new black and
29 white Laplacian movie?

30 Yes, the new black and white Laplacian movie has been labelled as Extend Data Mov.3 in the
31 Supplementary Information during the previous submission. We apologize that we did not
32 upload it as a separate file. In the new submission, all these three movies are uploaded separately.

33 Minor points

34 1. Lines 59-60, the ligand is negatively charged and as per IUPAC rules it should be 5-iodo-N-
35 (8'-quinolyl)-salicylaldiminate. This error is present throughout the manuscript.

36 We thank the referee for this correction. “5-iodo-N-(8'-quinolyl)-salicylaldiminate” is correct in
37 the main manuscript and SI, after checking the IUPAC nomenclature. The name of the dmit is also
38 updated as “1,3-dithiole-2-thione-4,5-dithiolate”.

39 2. Line 124, DAS is used but not explained.

40 Corrected.

41 3. Figure 1 remains more or less the same as before. Both myself and the other referee suggested
42 that arrows or similar indicators be used to show the changes that occur upon photoexcitation.

43 It appears that the referee was misled regarding Fig. 1b. In the resubmitted modification, we
44 revised the caption of Fig. 1b as “schematic of the structure in the high temperature phase”.
45 Therefore, it does not represent changes upon photoexcitation as mentioned in the first
46 submission. Instead, the changes resulting from photoexcitation, such as Fe-ligand expansion
47 and sliding displacement, are depicted in Fig. 4f. We apologize to the referee for the wrong
48 caption in the first manuscript.

49 Our rationale is to provide a clear depiction of the molecule's structure, comprising Fe(III)
50 centered SCO molecules with two iodine-substituted ligands and the $[\text{Ni}(\text{dmit})_2]^-$ anions,
51 arranged in a one dimensional zigzag molecular array with alternating face-to-face and side-
52 by-side manner in Fig. 1 as an introduction. Subsequently, we discuss structural changes upon
53 photoexcitation for later sections to prevent redundancy.

54 In the current manuscript, we have added more labels in Fig. 1b to emphasize the halogen bonds
55 between the cation and anion which are critical for the molecular structure.

56 4. A constant frustration for me in the ESI is having to move backwards and forwards to find
57 the figures referred to in the supplementary text. As this is the ESI, it would be far better to
58 place the figures near the text where they are referred to.

59 We are sorry for this frustration. In the new ESI, we have placed the figures near the text where
60 they are referred to. All the Extended Data Fig. numbers have been updated accordingly.

61 5. Extended figure 3, the caption use the word ‘blu’, when it should be blue.

62 Corrected.

63 6. In extended figure 12 it's very hard to read the labels for the atoms. I'd suggest that they are
64 made larger or a different colour is used to make them more visible. This in fact applies to the
65 new black and white movie they've created too.

66 We have updated the Extended Figure 12 as Extended Figure 13. To show the label clearly, we
67 improved the location of the label with larger size and different color. This change applies to
68 the new movie (Extended Data Mov. 3).

69 Overall, I am happy with the changes made by the authors and I think the manuscript now more
70 accurately reflects the results and clarifies the areas of confusion. Provided that they undertake
71 these revisions I would be happy to recommend the paper for publication in Nature
72 Communications.

73

74 Referee #2 (Remarks to the Author):

75 The paper is a new version of the paper by Yifeng Jiang et al, previously submitted to [REDACTED]
76 is about ultra-fast pump-probe study of photoinduced phenomena in a material made of a
77 photoactive spin crossover molecular unit, connected through halogen-bonds to (dmit)₂ units
78 exhibiting strong or weak dimerization. The authors used ultrafast optical and infra-red
79 spectroscopies and complementary ultrafast electron diffraction, to gain knowledge on ultrafast
80 and out-of-equilibrium photoinduced dynamics. The experimental data reveal ultrafast changes
81 within 500 fs and slower dynamics within 50 ps. In this new version of the paper, as well as
82 Supp. Info, the authors have considered the most important remarks and have improved the
83 presentation of the results. The analysis of the results and the discussion are more relevant and
84 rigorous in the present version, with several additional discussions about the method or the data
85 analysis, and the associated limitations, such as about the role of the volume change and the
86 halogen bonds. The extended data, especially the one about Halogen bond, make the discussion
87 stronger. Now it is clear why the two different timescales can be associated with two different
88 dynamics and how they couple. This work is very promising for exploring enhanced control of
89 multifunctional through halogen bond and it can attract a broad community of researchers.

90

91 The reviewer recommends publication of the paper in its present form, considering the minor
92 points below:

93 1- It is indicated line 111 "Fig. 2b shows an approximate match of all the transient spectra of
94 after +1 ps time delay with the characteristic of the formation of the HS state from the
95 temperature-dependent measurement."

96 However, few lines above it is indicated "OD of the HT phase is lower than that of the LT phase
97 at wavelengths shorter than $\lambda_0 = 530$ nm and higher at wavelengths longer than λ_0 in the visible
98 region" Therefore, it seems from Fig2b that it is only after 100 ps that OD transient spectra
99 represent an approximate match characteristic of the formation of the HS state."

100 I also found a typo: "transient spectra of after +1 ps time delay"

101 "transient spectra after +1 ps time delay"

102

103 We thank the referee for pointing out this. We agree that the OD transient spectra after +100 ps
104 has a better approximate match of the characteristics of the formation of the HT phase spectra.
105 We noticed our description in line 111 had been a misleading one. We changed "+1 ps" to "+100
106 ps" as suggested by referee. In addition, we changed "formation of the HS state" to "HT phase".
107 As written in the next paragraph, we assigned HS state formation as a very fast dynamics with
108 the time constant of 0.16 ps. After +100 ps, the system is after the sequential active of the dimer
109 softening process and it is in a more equilibrium state like the thermally-induced phase
110 transition. As shown in Extend data Fig 5 and Extended Data Fig. 6, the main difference
111 between transient spectra after +100 ps and spectra after +1 ps and +5 ps is two additional

112 relaxation components with 13.4 and 50.1 ps time constant, which might be related to dimer
113 softening of the $[\text{Ni}(\text{dmit})_2]^-$ anions.

114 Correction : (**Line 111-113**). Fig. 2b shows an approximate match of all the transient spectra
115 after +100 ps time delay with the characteristic of the HT phase from the temperature-dependent
116 measurement.

117 Reviewer #2 (Remarks on code availability):

118 Dear editor,

119 I'm sorry, I am not able to review the code. I have no electron data to run or test

120 Apologies for any inconvenience during the code test. Further details about the custom code
121 can be found in the Method and Code available sections. All necessary data, including UED
122 data and crystal information, has been uploaded to the "data" section of the capsule. You'll find
123 the custom codes in the "code" section, while the results, including figures and data generated
124 from the custom codes, are presented in the "results" section. The code has been validated, so
125 to test it, just run the "Reproducible Run."

REVIEWERS' COMMENTS

Reviewer #1 (Remarks to the Author):

I think that the authors have now adequately addressed all of my concerns and I would be pleased to see the paper accepted in Nature Communications

Reviewer #1 (Remarks on code availability):

I have quickly looked at the code but I am unqualified to judge whether or not the code is absolutely correct.